# The Importance of Subsurface Processes in Land Surface Modeling over a Temperate Region: An Analysis with SMAP, Cosmic Ray Neutron Sensing and Triple Collocation Analysis

Haojin Zhao [1,2,*], Carsten Montzka [1], Roland Baatz [1,3], Harry Vereecken [1,2,3] and Harrie-Jan Hendricks Franssen [1,2]

1   Agrosphere (IBG-3), Forschungszentrum Jülich, 52425 Jülich, Germany; c.montzka@fz-juelich.de (C.M.); r.baatz@fz-juelich.de (R.B.); h.vereecken@fz-juelich.de (H.V.); h.hendricks-franssen@fz-juelich.de (H.-J.H.F.)
2   Centre for High-Performance Scientific Computing in Terrestrial Systems, HPSC TerrSys, 52425 Jülich, Germany
3   Scientific Coordination Office, International Soil Modelling Consortium ISMC, 52425 Jülich, Germany
*   Correspondence: h.zhao@fz-juelich.de

**Abstract:** Land surface models (LSMs) simulate water and energy cycles at the atmosphere–soil interface, however, the physical processes in the subsurface are typically oversimplified and lateral water movement is neglected. Here, a cross-evaluation of land surface model results (with and without lateral flow processes), the National Aeronautics and Space Administration (NASA) Soil Moisture Active/Passive (SMAP) mission soil moisture product, and cosmic-ray neutron sensor (CRNS) measurements is carried out over a temperate climate region with cropland and forests over western Germany. Besides a traditional land surface model (the Community Land Model (CLM) version 3.5), a coupled land surface-subsurface model (CLM-ParFlow) is applied. Compared to CLM stand-alone simulations, the coupled CLM-ParFlow model considered both vertical and lateral water movement. In addition to standard validation metrics, a triple collocation (TC) analysis has been performed to help understanding the random error variances of different soil moisture datasets. In this study, it is found that the three soil moisture datasets are consistent. The coupled and uncoupled model simulations were evaluated at CRNS sites and the coupled model simulations showed less bias than the CLM-standalone model ($-0.02 \text{ cm}^3 \text{ cm}^{-3}$ vs. $0.07 \text{ cm}^3 \text{ cm}^{-3}$), similar random errors, but a slightly smaller correlation with the measurements (0.67 vs. 0.71). The TC-analysis showed that CLM-ParFlow reproduced better soil moisture dynamics than CLM stand alone and with a higher signal-to-noise ratio. This suggests that the representation of subsurface physics is of major importance in land surface modeling and that coupled land surface-subsurface modeling is of high interest.

**Keywords:** soil moisture; cosmic-ray neutron sensors (CRNS); SMAP; land surface model; subsurface; triple collocation (TC)

## 1. Introduction

Soil moisture exerts an important control on the water and energy cycles in the atmosphere-land surface-subsurface continuum. Therefore, improving soil moisture estimation is beneficial for understanding the partitioning of water and energy fluxes. Soil moisture variability in the unsaturated zone is affected by water exchanges between the unsaturated zone, atmosphere, and groundwater. Studies have been focused on the effects of groundwater dynamics on land surface processes, showing the role of the groundwater in the water and energy cycles [1,2]. Results indicate that groundwater has little impact on soil moisture in deep groundwater regions, however, in districts with shallow groundwater—such as wetlands and river valleys—groundwater can become a major source of soil water [3–5]. The groundwater table depths and hydraulic gradients between

saturated and unsaturated soils can cause capillary rise and make groundwater a constant water supply, leading to altered runoff and evaporation rates [6,7]. The land surface energy balance is affected by soil moisture states and land surface temperature [8–10].

Despite the importance of subsurface flow, most current land surface models (LSMs) neglect the lateral flow between grid cells or at sub-grid scales and only consider the water exchange in the vertical direction. Such models include the Variable Infiltration Capacity model (VIC), Interaction Soil Biosphere Atmosphere (ISBA) surface scheme, Noah Land Surface Model, and Community Land Surface Model (CLM). Recently, some works have considered the role of lateral subsurface flow and developed three dimensional hydrological models coupled with LSMs, such as CATHY (CATchment HYdrology), NoahMP [11], VIC-MD (MODFLOW) [12], and CLM-ParFlow [13]. These models aim at simulating the subsurface water and energy cycles more realistically than uncoupled models. However, model predictions are affected by errors given the uncertainty of the many required input parameters, e.g., atmospheric forcing, soil texture, and vegetation properties [14]. Precise soil hydraulic parameter data and land cover type information are hard to obtain as most areas lack sufficient land surveys. Also, the model structure and further assumptions influence the simulation performance. A large number of parameters used in land surface models are hard-coded as constants, although they are calculated by linear regression from preliminary studies using a limited amount of data and known to be uncertain [15].

Soil moisture can be measured via automated techniques such as gravimetric methods, nuclear techniques (such as neutron scattering, Gamma ray attenuation), electromagnetic methods (e.g., the time domain reflectometry (TDR) and the frequency domain reflectometry (FDR)), tensiometers and hygrometry [16–18]. However, most sensors monitor soil moisture at point scales (radius less than 1 m) only. As soil moisture is very heterogeneous in space and time, one usually needs to collect data from multiple locations in a specific area [19]. To reduce the scale gap to remote sensing products and modeling results and to obtain area-averaged soil moisture, a new technology has emerged with the Cosmic Ray Neutron Sensor (CRNS). It measures neutron count intensity and determines soil moisture in a non-invasive and continuous way [20]. The omnipresent cosmic radiation produces neutrons that interact with atmosphere and ground. These secondary neutrons include fast neutrons, that are generated by collisions between high-energy neutrons and nuclei. Fast neutrons are easily moderated by hydrogen atoms. As soil is the main source of hydrogen, thus, the variations of fast neutrons are strongly related to soil moisture changes. The process of moderation can be captured and counted by cosmic ray neutron sensors [21,22]. The large spatial footprint makes it suitable for agricultural water resources management and remote sensing product validation [23].

Two innovative satellite missions that include L-band (1200–1400 MHz) passive microwave systems have already been launched, including SMOS (Soil Moisture and Ocean Salinity mission, launched in November 2009) [24] and SMAP (Soil Moisture Active Passive mission, launched in January 2015 and starting operations in April 2015) [25]. Both SMOS and SMAP aim to measure soil moisture globally. After SMAP radar stopped operation in July 2015, an enhanced product was developed to give high-resolution observations at 9 km resolution. This enhanced product was evaluated by comparing it with long-term in situ soil moisture data [26,27]. It was found that the average ubRMSE (unbiased Root Mean Square Error) in L2_SM_P_E (Level 2 Enhanced Passive Soil Moisture) product and L3_SM_P_E (Level 3 Enhanced Passive Soil Moisture) product are between $-0.040 \sim 0.055$ cm$^3$ cm$^{-3}$ [27–29], which barely meets the accuracy requirements of the SMAP mission. A number of previous studies have compared SMAP products and/or model simulations with in-situ observations. EI Hajj et al. [30] evaluated SMAP soil moisture products at sites in Southwestern France and found that the average bias over stations was about $-0.032$ cm$^3$ cm$^{-3}$, indicating SMAP moderately underestimates the soil moisture compared to in situ observations. Walker et al. [31] compared SMAP soil moisture with validation sites in the South Fork River watershed in Iowa, U.S., and found that the bias could be up to $-0.04$ cm$^3$ cm$^{-3}$ in early-spring and late fall and improve to $-0.02$ cm$^3$ cm$^{-3}$ in the summer time. At global scale, a study indicated that SMAP

shows a dry bias [32]. Recently, a new COSMOS network is developed to provide a unified, standardized, publicly available, traceable, and objective validation procedure that is operational in ISMN (https://www.geo.tuwien.ac.at/insitu/data_viewer/ accessed on 1 August 2021) and the QA4SM online validation service for soil moisture products (https://qa4sm.eu/ accessed on 1 August 2021) [33]. Here, soil moisture from ERA5, ERA5-Land, and GLDAS are provided for validation, but subsurface processes are not considered. As mentioned, SMAP was only compared to soil moisture simulated by stand-alone land surface models that have a rather simple subsurface structure and do not consider the role of groundwater and lateral flows [34–36], which could lead to systematic deviations. Studies show that soil moisture is overestimated when the model neglects the impact of topography [37,38]. Also, previous studies focused mostly on large scales and simulations at a coarser spatial resolution [23,39,40].

To find relative error estimates for different products, the Triple Collocation (TC) method is used, which is an error magnitude estimation approach for intercomparison among three or more independent observed or modeled datasets [41]. The method was firstly used for ocean wind studies and then developed and widely used in other areas, such as land surface hydrology [42–47]. It has been proven to be a useful tool to understand the random error variances of remote sensing time series. It assumes one of these datasets as reference to relative rescaling, and further assumes truth-error orthogonality and zero error cross correlation between datasets to obtain a bias-free TC analysis [46–48]. Here, we present simulated soil moisture by CLM and the CLM-ParFlow coupled model and compared it with CRNS observations and SMAP enhanced soil moisture datasets using TA method. Compared to the CLM model, CLM-ParFlow considers three-dimensional water flow in the subsurface (soil and aquifer) and a two-dimensional overland flow module. It is investigated whether a better subsurface representation can improve soil moisture estimates. The research area has various land use types and various pedological, hydrological, and hydrogeological site conditions were observed. These small-scale hydrological processes can provide insights for large scale modeling. This research aims to understand the performance and limitations of the subsurface role in land surface model simulations, which might recall the need to consider hydrogeology (including complex 3D geology and critical parameters like hydraulic conductivity and storage coefficients) in soil-vegetation-atmosphere processes, and to obtain finally more accurate soil moisture and groundwater level data.

## 2. Materials and Methods

### 2.1. Study Area

The study area is located in central Europe encompassing parts of North Rhine-Westphalia and Rhineland-Palatinate in western Germany and parts of Belgium, the Netherlands and Luxemburg, covering an area of 150 × 150 km (Figure 1). This region has a sub-Atlantic oceanic climate. Summers are mild, while winters are humid and relatively mild. The average monthly temperatures are highest in July (18 °C) and lowest in January (3 °C). There is a large spatial variability in precipitation due to topography. In the rather flat Lower Rhine area in the North of the study region, the yearly precipitation is between 600–900 mm [49]. In the southern low mountain ranges the average yearly precipitation is locally 1600 mm in the Bergisches Land (South East) and 1300 mm in the Eifel (South West) [50]. The rainfall is frequent and evenly distributed over the seasons. The elevation in this area ranges from the plains at around 14 m a.s.l. to the mountainous areas at around 735 m.

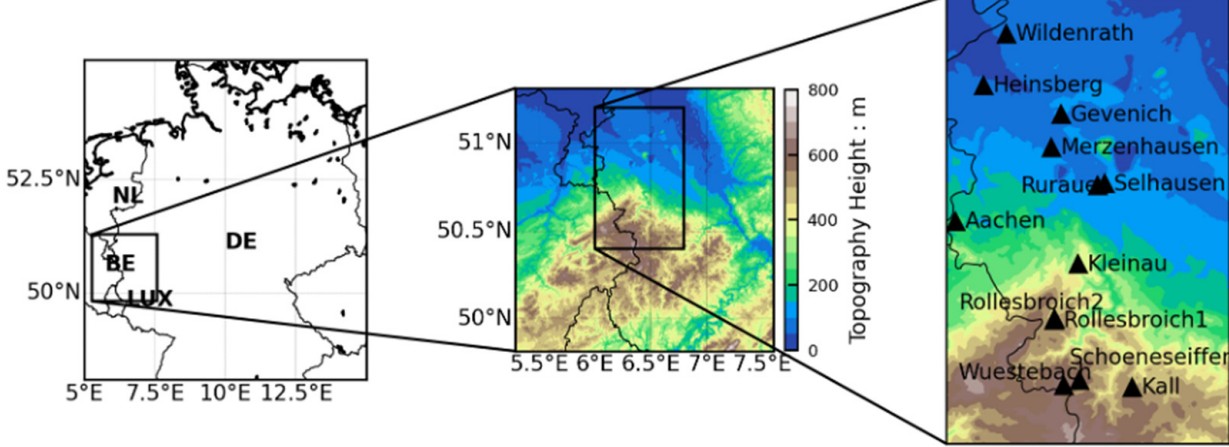

**Figure 1.** The research area and 13 CRNS locations (denoted as black triangles).

The land use is a mixture of agriculture area, forests, urban and rural areas, water, grassland, industry, and mining [2]. The dominant land use is agriculture, covering more than 60% of the area (shown in Figure 2). The major crops are winter wheat, maize, and sugar beet. Forests cover nearly 20 % of this area and are mainly located in the south, i.e., Eifel, Bergisches Land, and Sauerland. The dominant soil textures are sandy loam, loam, and clay loam and our simulations are based on the FAO/UNESCO Soil Map [51]. Sandy soils with low water holding capacities are mainly located in the Northwest region.

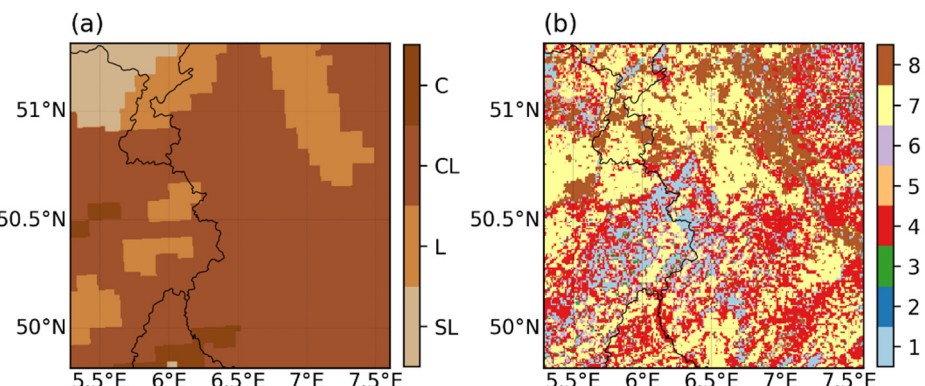

**Figure 2.** Map of the North Rhine-Westphalia (NRW) domain for (**a**) soil type and (**b**) plant functional types. The soil types are divided into sandy loam (SL), loam (L), clay loam (CL) and clay (C). The PFTs are defined as: 1–needleleaf evergreen tree (NET); 2–needleleaf deciduous tree (NDT); 3–broadleaf evergreen tree (BET); 4–broadleaf deciduous tree (BDT); 5–broadleaf deciduous shrub (BDS); 6–grassland (GRASS); 7–crop (CROP), 8–barren soil (BARE).

This research area has three solid rock areas of regional importance [52]. The largest part is Palaeozoic rocks located in the southern part and east of the river Rhine, having the dominant aquifer typology 'schist and shales' that includes folded and partly metamorphosed clastic sedimentary rocks. This part has a low hydraulic conductivity. The north-eastern part is mainly occupied by unconsolidated rocks of Cenozoic era, including alternating sequences of clastic sedimentary rocks horizontally. The hydraulic conductivity varies a lot depending on the combination system. In the western part, the dominant aquifer rocks are mostly consolidated, weakly permeable material from Mesozoic.

Soil moisture validation activities have been performed in this area before based on in situ and CRNS data [23,53–56] as well as simulation and data assimilation experiments [57–60].

*2.2. Data*

2.2.1. Land Surface Modeling

Soil moisture was simulated by the land surface model CLM (community land model) and the coupled land surface–subsurface model CLM–ParFlow. The CLM model was developed by the National Center for Atmosphere Research (NCAR). In the CLM 3.5 model, the soil is discretized into 10 unevenly distributed soil layers (see Table 1). Soil hydraulic properties are estimated internally from soil texture (sand fraction and clay fraction) using pedo-transfer functions according to Clapp and Hornberger [61] and Cosby et al. [62]. A simplified Richards equation is used in CLM to calculate the vertical water movement in the unsaturated zone

$$\frac{\partial \theta}{\partial t} = \frac{\partial}{\partial z}\left[K \times \frac{\partial(\varphi - z)}{\partial z}\right] + Q \tag{1}$$

where $\frac{\partial \theta}{\partial t}$ is the soil moisture (cm$^3$ cm$^{-3}$) change over time, $K$ represents the saturated hydraulic conductivity (m/s), $\varphi$ is the pressure head (unit is defined as length (L)), $z$ as the vertical coordinate (L) and $Q$ is the source/sink term (i.e., the soil water removed due to evaporation). A main limitation of CLM 3.5 is that lateral flows are not considered, and groundwater is not represented, although groundwater can strongly influence soil moisture conditions.

**Table 1.** Soil layer depth in CLM model.

| Soil Layer | 1 | 2 | 3 | 4 | 5 | 6 | 7 | 8 | 9 | 10 |
|---|---|---|---|---|---|---|---|---|---|---|
| Depth (m) | 0.010 | 0.035 | 0.075 | 0.135 | 0.235 | 0.400 | 0.650 | 1.050 | 1.650 | 2.500 |

ParFlow [63] is a numerical, integrated hydrological model that simulates subsurface groundwater flow and water flow in soils, as well as overland flow. Both retention and relative permeability curves are represented by the van Genuchten relationships [64]. ParFlow does not include land surface process (e.g., evaporation), nor does it have a parameterization scheme for frozen soil and ice processes. In addition to the DEM dataset used in CLM, the topographic slopes need to be specified for ParFlow. When coupled with CLM, ParFlow replaces the one-dimensional CLM soil moisture characterization by the three-dimensional approach in CLM-ParFlow, considering the redistribution of soil moisture, and integrating vertical and lateral flow of groundwater and surface water.

$$S_s S_w \frac{\partial \varphi}{\partial t} + \phi \frac{\partial S_w(\varphi)}{\partial t} + Kk_r(\varphi) \cdot \nabla(\varphi - z) = Q \tag{2}$$

In this equation, $S_s$ is the specific storage coefficient (L$^{-1}$), $S_w$ is the relative saturation, $\phi$ is the porosity, $k_r$ is the relative permeability. The subsurface is discretized into 30 layers, with 10 vertically layers near the surface (2–100 cm) and 20 constant levels (135 cm depth) that reach up to 30 m below the surface. The physically based coupled model (CLM-ParFlow) can better simulate the role of groundwater in terrestrial systems, and the interaction between surface water and subsurface [13,63].

To represent the high spatial heterogeneity of the land surface, the simulation domain was discretized into grid cells of 500 × 500 m. The plant functional types (PFTs) were based on MODIS land cover data. The hilly areas are mostly covered by broadleaf forest and needleleaf forest, while the other flat regions are covered by crops and urban areas. Soil texture information was taken from the FAO/UNESCO Soil Map [51] with the scale of 1: 5,000,000 (see Figure 2). Most of the model domain is dominated by clay-loam (35% clay, 35% sand, 30% silt). Sandy loam (10% clay, 65% sand, 25% silt) and loam (20% clay, 40% sand, 40% silt) are dominant in the north-western part. Clay (45% clay, 15% sand, 40% silt) is found in the northwestern corner of the domain. The van Genuchten water retention

parameters and the hydraulic conductivity used in CLM-ParFlow are calculated by the Rosetta pedo-transfer function [65], summarized in Table 2.

**Table 2.** Subsurface hydraulic properties used in CLM-ParFlow simulations.

| Soil Type | Clay | Clay Loam | Loam | Sandy Loam |
|---|---|---|---|---|
| $K$ (m hr$^{-1}$) | 0.0062 | 0.0034 | 0.0050 | 0.0158 |
| $\alpha$ (m$^{-1}$) | 2.1 | 2.1 | 2.0 | 2.7 |
| n | 2.0 | 2.0 | 2.0 | 2.0 |
| $\theta_s$[1] | 0.4701 | 0.4449 | 0.4386 | 0.4071 |
| $\theta_r$[1] | 0.21 | 0.17 | 0.15 | 0.1 |

[1] $\theta_s$ saturated water content (cm$^3$ cm$^{-3}$); $\theta_r$ residual water content (cm$^3$ cm$^{-3}$).

To drive the model, the high-resolution reanalysis dataset (COSMO-REA6) [66] was used as meteorological forcing. This dataset covers the period 1995–2020 and is continuously supported by DWD (Deutscher Wetterdienst, German Meteorological Service). It uses ERA-Interim data as a boundary condition and is generated by assimilating observed meteorological data into the atmospheric model COSMO (Consortium for Small-scale Modeling) [67]. The dataset comprises air temperature, precipitation, humidity, incoming shortwave and longwave radiation. A two-year spin-up period was applied for CLM and the initial conditions for CLM-ParFlow were taken from previous studies [2,59]. Both simulations with CLM and CLM-ParFlow started from a near equilibrium condition. Thus, different spin up treatments do not have influence on results. In total, a period of two years (2017–2018) was simulated with a time step of 3600 s. In case of convergence issues, the time steps are reduced until convergence can be achieved.

Simulated soil moisture for the upper 5 cm layer (SM$_{5cm}$) and the upper 20 cm (SM$_{20cm}$) were estimated by linearly combining simulated output for different model layers (H20SOI$_i$, where $i$ is the index denoting the soil layer).

$$SM_{5cm} = 0.14 \times H20SOI_1 + 0.56 \times H20SOI_2 + 0.30 \times H20SOI_3 \tag{3}$$

$$SM_{20cm} = 0.0165 \times H20SOI_1 + 0.0651 \times H20SOI_2 + 0.1451 \times H20SOI_3 + 0.2770 \times H20SOI_4 + 0.4963 \times H20SOI_5 \tag{4}$$

### 2.2.2. CRNS Observations

The CRNS is an emerging technology to monitor soil moisture at the intermediate scale [20,22]. The measured neutron count intensity provides an estimate of soil moisture content for a radius of around 240 m, at sea level and dry bare soil conditions. The radius is a function of air density, air humidity and vegetation density [68]. The penetration depth of the CRNS measurements varies from 15 cm (wet soils) to 55 cm (dry soils) [69]. The neutron count intensity is mainly sensitive to the number of hydrogen atoms in the soil, but is also influenced by changes in atmospheric pressure, vapor pressure, and incoming cosmic radiation. These factors are considered in the standard correction process [69,70].

Several studies have been conducted to investigate the accuracy of the CRNS measurements and found that CRNS provides reliable soil moisture estimates when calibrated properly [71–73]. Bogena [74] found that even for a densely vegetated and wet site, the RMSE of daily soil moisture estimated by CRNS is only 0.03 cm$^3$ cm$^{-3}$. In our work, 13 CRNS stations are used to evaluate SMAP and soil moisture model products (see Figure 1). The datasets are collected in the context of the TERENO project [75] and passed quality assurance procedures. We acknowledge that the effective depth of CRNS is dependent on soil moisture, and also on the depth of the calibration dataset. Our CRNS calibration and validation used individual support volumes of samples from 5, 20, and 50 cm based on the gravimetric method. We calculate the penetration depth based on

previous study [76] and found that the effective penetration depth is mostly between 15–30 cm in our research area and period. The mean and median of penetration depth are 20.3 cm and 18.6 cm respectively. Hence, we assumed that the CRNS has an effective penetration depth of 20 cm. Table 3 provides more information about the CRNS stations.

**Table 3.** Coordinates, altitude (m), average annual precipitation (mm y$^{-1}$), land use type information [60] for 13 sites, adapted with permission from ref. [60]. Copyright 2017 by Roland Baa.tz.

| Name | Latitude | Longitude | Altitude | Precip. | Land Use | Clay % | Sand % | Bulk g cm$^{-3}$ |
|---|---|---|---|---|---|---|---|---|
| Merzenhausen | 50.9303 | 6.29747 | 94 | 825 | crop | 22 | 21 | 1.39 |
| Aachen | 50.7986 | 6.02472 | 232 | 952 | crop | 23 | 22 | 1.20 |
| Selhausen | 50.8659 | 6.44719 | – | – | crop | 24 | 16 | 1.26 |
| Heinsberg | 51.0411 | 6.10424 | 57 | 814 | grassland, crop | 19 | 18 | 1.27 |
| Wüstebach | 50.5049 | 6.33092 | 605 | 1401 | spruce | 23 | 19 | 0.83 |
| Gevenich | 50.9892 | 6.32355 | 108 | 884 | crop | 20 | 22 | 1.31 |
| Rollesbroich1 | 50.6219 | 6.30424 | 515 | 1307 | grassland | 23 | 22 | 1.09 |
| Rollesbroich2 | 50.6242 | 6.30514 | – | – | grassland | - | - | 1.09 |
| Ruraue | 50.8623 | 6.42734 | 102 | 743 | grassland | 26 | 19 | 1.12 |
| Wildenrath | 51.1327 | 6.16918 | 76 | 856 | needleleaf | 12 | 65 | 1.15 |
| Kall | 50.5013 | 6.52645 | 504 | 935 | grassland | 22 | 20 | 1.31 |
| Schoeneseiffen | 50.5149 | 6.37559 | – | – | grassland | 24 | 16 | 1.11 |
| Kleinhau | 50.7224 | 6.37204 | – | – | grassland | 25 | 15 | 1.12 |

### 2.2.3. SMAP Enhanced Soil Moisture Product

SMAP provides soil moisture observations of the top 5 cm of the soil and thaw/freeze states derived from the passive microwave brightness temperature (BT). BT is recorded by a conically-scanning antenna beam at L-band with a 40° incidence angle. This results in a −3 dB antenna footprint of 40 km. To enhance the resolution of the typically 36 km SMAP radiometer data posting, the Backus-Gilbert optimal interpolation technique is used to interpolate the multiple scans of a single location. It makes most use of the available information and provides a better representation of the original data [26].

In this study the L3_SM_P_E product (version 4) was used, which provides soil moisture on a 9 km EASE2 (updated Equal-Area Scalable Earth-2) grid (National Snow and Ice Data Center NSIDC, https://nsidc.org/data/smap/smap-data.html, accessed on 1 August 2021). The soil moisture baseline retrieval algorithm in L3_SM_P_E product is performed by the vertical polarization single channel algorithm (SCA-V) (https://nsidc.org/data/smap/technical-references, accessed on 1 August 2021). The L3_SM_P_E product is provided in the form of global daily datasets, including soil moisture measured for the 6:00 a.m. (descending) and 6:00 p.m. (ascending) orbit. Here, the soil moisture daily value is calculated by taking the average of the two datasets. To eliminate the non-high-quality pixels, the surface and quality flags are used (retrieval_qual_flag and surface_flag).

### 2.3. Methods

#### 2.3.1. Data Processing

Both 2017 (normal year) and 2018 (dry year) were selected to be evaluated for all datasets. In order to avoid unreliable soil moisture observations during frozen conditions and snow cover, the winter period (December, January, and February) was excluded.

The model simulations and SMAP product are sampled onto the SMAP grid by nearest neighbor (NN) search. Compared to area-wide spaceborne observation and model simulation results, the CRNS stations are quite sparse. As the spatial coverage of measurements by a CRNS is close to the model grid size in this work, CRNS observations are compared to a complete cell of the CLM, CLM-ParFlow, and SMAP grid containing the coordinates of the CRNS stations.

The SMAP product and the model grids are at different spatial resolutions (9 km vs. 500 m) and comparison between SMAP soil moisture and modeled soil moisture is made at both resolutions. The fine resolution (model grid) provides a detailed assessment of the spatial variability, and the coarse resolution (SMAP grid) gives a smoothed representation excluding local noise. The comparison at the 9 km resolution is made by taking the arithmetic average of simulated soil water content data on the 500 m grid to get modeled values at the 9 km resolution (upscaling). The comparison at the 500 m resolution is made by downscaling the SMAP data, using the nearest source to destination to remap from 9 km resolution to 0.5 km resolution. In this case, for each model grid cell, it takes the value from the nearest SMAP grid cell. This is done by ESMF (Earth System Modeling Framework) regridding function in NCL (NCAR Command Language).

### 2.3.2. Standard Evaluation Metrics

Statistical performance was evaluated according to Good Practices Guidelines [19,77], including bias, Root Mean Square Difference (RMSD), and unbiased Root Mean Square Error (ubRMSD). The bias is given by

$$\text{bias} = \frac{1}{n}\sum_{i=1}^{n}(X_i - X_{i,ref}) \tag{5}$$

where $X_i$ represents a simulated or remotely sensed product, and $X_{i,ref}$ is the referenced soil moisture dataset. The sample size is $n$.

The Root Mean Square Deviation (RMSD) is given by

$$\text{RMSD} = \sqrt{\frac{1}{n}\sum_{i=1}^{n}(X_i - X_{i,\text{ref}})^2} \tag{6}$$

and the unbiased root mean squared difference (ubRMSD) is

$$\text{ubRMSD} = \sqrt{\text{RMSD}^2 - \text{bias}^2} \tag{7}$$

The correlation between different soil moisture datasets ($X$ and $Y$) was calculated using the Pearson correlation coefficient ($r_{X,Y}$). Here, the focus is on the dynamics of the different time series rather than absolute soil moisture values, considering the fundamental scale mismatch between the CRNS, SMAP, and LSM model grid, which cannot be adequately considered without complex scaling functions. The ubRMSD and $r_{X,Y}$ screen out the influence of bias between different amplitudes of soil moisture variation [78].

$$r_{X,Y} = \frac{cov(X,Y)}{\sigma_X \sigma_Y} \tag{8}$$

where $cov(X,Y)$ represents the covariance between two datasets, $\sigma_X$ and $\sigma_Y$ are the standard deviations for dataset $X$ and $Y$.

### 2.3.3. Triple Collocation

The method is based on the assumption that a measurement system can be understood as being composed of an additive systematic error ($\alpha_i$), multiplicative systematic error ($\beta_i$), additive zero-mean random error ($\varepsilon_i$), and $\theta$ (the underlying truth value) [45]

$$i = \alpha_i + \beta_i \theta + \varepsilon_i \ \ i \in [X, Y, Z] \tag{9}$$

$X$, $Y$, and $Z$ denote the time series of soil moisture products to be compared. Here, the assumption is that the errors are uncorrelated with each other ($cov(\varepsilon_i, \varepsilon_j) = 0, i \neq j$) and with $\theta$ ($cov(\varepsilon_i, \theta) = 0$). It applies for both reference datasets and soil moisture products to be evaluated. Given that all observation data has errors, in this study, we assumed

that $X$ is the reliable CRNS dataset which is well calibrated and without multiplicative (second order) error, $Z$ is the SMAP L3 enhanced product, and $Y$ denotes the soil moisture in the upper 5 cm of CLM and CLM-ParFlow simulations. The $\beta_x$ is set to 1, and other scaling factors are calculated according equations 10-12 to eliminate the scale differences of different products. This assumption was also made in other studies [45,77,79].

$$\beta_x = 1 \tag{10}$$

$$\beta_y = \frac{cov(X,Z)}{cov(Y,Z)} \tag{11}$$

$$\beta_z = \frac{cov(X,Y)}{cov(Z,Y)} \tag{12}$$

The scaling factor can be regarded as a form of regression, by taking a third variable as a tool to resolve the relationship between the two variables [80]. It can be used to describe the soil moisture sensitivities. The unscaled error variances which represent the variance of the scaled white noise of each product can be solved by

$$\sigma_{\varepsilon X}^2 = \left| var(X) - \frac{cov(X,Y)cov(X,Z)}{cov(Y,Z)} \right| \tag{13}$$

$$\sigma_{\varepsilon Y}^2 = \left| var(Y) - \frac{cov(Y,X)cov(Y,Z)}{cov(X,Z)} \right| \tag{14}$$

$$\sigma_{\varepsilon Z}^2 = \left| var(Z) - \frac{cov(Z,Y)cov(Z,X)}{cov(Y,X)} \right| \tag{15}$$

In addition, the signal-to-noise ratio (SNRs) is calculated, which provides a more clear representation of the ratio between soil moisture and uncertainty magnitude [77]. It is a combination of several information sources, including the sensitivity of the measurement system, the variability of the true value ($\theta^2$) and the variability of the random error ($\varepsilon^2$) [44]. SNRs is not expressed as normalized between 0 and 1, but is often calculated and linearized by converting into decibel (dB) units

$$SNR_X = -10\log\left(\left| \frac{var(X)cov(Y,Z)}{cov(X,Y)cov(X,Z)} - 1 \right|\right) \tag{16}$$

$$SNR_Y = -10\log\left(\left| \frac{var(Y)cov(X,Z)}{cov(Y,X)cov(Y,Z)} - 1 \right|\right) \tag{17}$$

$$SNR_Z = -10\log\left(\left| \frac{var(Z)cov(X,Y)}{cov(Z,X)cov(Z,Y)} - 1 \right|\right) \tag{18}$$

## 3. Results and Discussions

### 3.1. Agreement between Spaceborne and In Situ Observations

Table 4 shows that the SMAP L3_SM_P_E product and CRNS observations have in general a relatively good agreement. The detailed time series of soil moisture from SMAP, CRNS and LSMs at 13 sites are provided in Appendix A (Figure A1). The SMAP product and CRNS show on average not large systematic differences, while previous studies [23,39] found that SMAP tends to underestimate soil moisture compared to CRNS measurements at most sites. In terms of average $r$ value, the SMAP is relatively well correlated to CRNS, ranging from 0.653 to 0.825, except for Ruraue station ($r = 0.452$). The Ruraue station is located near the Rur river, and part of the closest SMAP pixels contain some amount of open fresh water. The presence of open water introduces a soil moisture bias due to the lower brightness temperature for the grid cell. This may partly explain the low $r$ for the Ruraue station.

**Table 4.** Comparison metrics for the SMAP L3_SM_E_P product compared to CRNS.

| Name | Bias: $cm^3$ $cm^{-3}$ | RMSD: $cm^3$ $cm^{-3}$ | ubRMSD: $cm^3$ $cm^{-3}$ | $r$ |
|---|---|---|---|---|
| Merzenhausen | 0.076 | 0.096 | 0.059 | 0.674 |
| Aachen | −0.003 | 0.049 | 0.049 | 0.768 |
| Selhausen | 0.031 | 0.066 | 0.059 | 0.653 |
| Heinsberg | 0.070 | 0.091 | 0.057 | 0.668 |
| Wüstebach | −0.120 | 0.133 | 0.057 | 0.752 |
| Gevenich | 0.067 | 0.088 | 0.058 | 0.684 |
| Rollesbroich1 | −0.023 | 0.060 | 0.055 | 0.741 |
| Rollesbroich2 | −0.053 | 0.077 | 0.055 | 0.708 |
| Ruraue | 0.030 | 0.080 | 0.075 | 0.452 |
| Wildenrath | 0.133 | 0.143 | 0.053 | 0.654 |
| Kall | −0.072 | 0.086 | 0.047 | 0.825 |
| Schoeneseiffen | −0.055 | 0.079 | 0.056 | 0.718 |
| Kleinhau | −0.018 | 0.051 | 0.048 | 0.789 |
| Average | 0.005 | 0.085 | 0.056 | 0.699 |

However, it should be noted that local differences are large. One reason is that the CRNS footprint is still much smaller than a 9 km SMAP-pixel. This spatial mismatch could lead to differing soil hydraulic parameters and land cover between CRNS and SMAP observations (see Tables 3 and 5) causing lower correlation between the two soil moisture datasets. It is also important to consider that the satellite observations are negatively impacted by high vegetation density, topography, frozen soil, snow cover, and volume scattering effect in case of low soil moisture content. The retrieval under dense forest is challenging or impossible since the recorded signal (brightness temperature) originates to a large extend from canopy instead of the soil microwave emissions [81,82]. When vegetation density increases, the impact of soil moisture on changes in ground emissivity becomes invisible, hence, the contribution of the ground is less than from the canopy. A large bias is observed in Wildenrath, where the land cover type is needleleaf (forest). Moreover, topography as well as surface roughness increases the surface area and alters the total microwave emission. In addition, it is also found that areas with complex topography are prone to shadowing and adjacency effects [83,84].

It should be noted that the soil organic matter content is high at the Wüstebach site, and there is a large difference of the bulk density between ancillary data (1.3 g $cm^{-3}$) and actual observed value (0.83 g $cm^{-3}$). In the single channel algorithm used by SMAP to retrieve soil moisture, the dielectric mixing model plays an important role in describing the relationship between soil moisture and microwave emissivity. A recent study has found that high levels of organic matter decrease the microwave effective dielectric constant and therefore cause higher brightness temperature for a particular soil moisture content [85].

*3.2. Comparison of Model Simulation and CRNS Measurements*

The 500 m modeled soil moisture was compared with CRNS measurements, assuming a measurement depth of 20 cm. Bias, RMSD, ubRMSD, and correlation coefficient were calculated for 13 stations and over three seasons (see Table 6 and Figure 3). The CLM simulations tend to overestimate soil moisture with a bias of 0.070 $cm^3$ $cm^{-3}$ and CLM-ParFlow has a slight dry bias of −0.021 $cm^3$ $cm^{-3}$. The large wet bias of the models for Wildenrath is most likely related to soil texture. Soil moisture decreases faster in sand than in finer textured soil. The sand content is high (up to 65%), and the models seem to have a too low hydraulic conductivity for this site. CLM-ParFlow shows a large bias at sites located at a higher elevation, such as Wüstebach, Rollesbroich, and Schoeneseiffen sites.

**Table 5.** Ancillary datasets used in the SMAP soil moisture retrieval algorithm. Notice that altitudes differ from Table 1 for the sites because here we list the altitudes used by SMAP.

| Name | DEM (m) | IGBP | Clay% | Sand% | Bulk Density (g cm$^{-3}$) |
|---|---|---|---|---|---|
| Merzenhausen | 79 | 12: Croplands | 21 | 39 | 1.40 |
| Aachen | 209 | 12: Croplands | 22 | 41 | 1.40 |
| Selhausen | 105 | 13: Urban and built-up lands | 23 | 37 | 1.40 |
| Heinsberg | 45 | 13: Urban and built-up lands | 21 | 39 | 1.40 |
| Wüstebach | 610 | 1: Evergreen needleleaf forests | 20 | 42 | 1.30 |
| Gevenich | 99 | 12: Croplands | 22 | 41 | 1.40 |
| Rollesbroich1 | 520 | 14: Cropland /natural vegetation mosaics | 20 | 42 | 1.30 |
| Rollesbroich2 | 520 | 14: Cropland /natural vegetation mosaics | 20 | 42 | 1.30 |
| Ruraue | 98 | 12: Croplands | 22 | 39 | 1.40 |
| Wildenrath | 79 | 5: Mixed forests | 22 | 41 | 1.40 |
| Kall | 510 | 14: Cropland /natural vegetation mosaics | 20 | 40 | 1.30 |
| Schoeneseiffen | 567 | 5: Mixed forests | 20 | 42 | 1.30 |
| Kleinhau | 347 | 5: Mixed forests | 20 | 42 | 1.30 |

**Table 6.** Comparison metrics between model simulations and in-situ observations.

| Name | CLM Simulations | | | | CLM-ParFlow Simulations | | | |
|---|---|---|---|---|---|---|---|---|
| | Bias: cm$^3$ cm$^{-3}$ | RMSD: cm$^3$ cm$^{-3}$ | ubRMSD: cm$^3$ cm$^{-3}$ | *r* | Bias: cm$^3$ cm$^{-3}$ | RMSD: cm$^3$ cm$^{-3}$ | ubRMSD: cm$^3$ cm$^{-3}$ | *r* |
| Merzenhausen | 0.108 | 0.136 | 0.050 | 0.711 | 0.045 | 0.105 | 0.094 | 0.414 |
| Aachen | 0.126 | 0.058 | 0.047 | 0.782 | −0.106 | 0.115 | 0.045 | 0.756 |
| Selhausen | 0.035 | 0.127 | 0.063 | 0.664 | 0.131 | 0.141 | 0.050 | 0.725 |
| Heinsberg | 0.111 | 0.088 | 0.049 | 0.785 | 0.005 | 0.083 | 0.083 | 0.365 |
| Wüstebach | 0.073 | 0.079 | 0.052 | 0.665 | −0.169 | 0.175 | 0.047 | 0.628 |
| Gevenich | −0.060 | 0.160 | 0.062 | 0.615 | 0.002 | 0.065 | 0.065 | 0.574 |
| Rollesbroich1 | 0.148 | 0.071 | 0.062 | 0.726 | −0.091 | 0.104 | 0.051 | 0.766 |
| Rollesbroich2 | 0.036 | 0.070 | 0.068 | 0.733 | −0.112 | 0.126 | 0.056 | 0.751 |
| Ruraue | 0.016 | 0.105 | 0.054 | 0.821 | 0.068 | 0.087 | 0.053 | 0.770 |
| Wildenrath | 0.090 | 0.186 | 0.039 | 0.755 | 0.078 | 0.083 | 0.029 | 0.800 |
| Kall | 0.182 | 0.079 | 0.079 | 0.576 | 0.007 | 0.075 | 0.075 | 0.570 |
| Schoeneseiffen | 0.008 | 0.077 | 0.070 | 0.778 | −0.088 | 0.101 | 0.050 | 0.805 |
| Kleinhau | 0.031 | 0.103 | 0.076 | 0.675 | −0.049 | 0.078 | 0.061 | 0.720 |
| Average | 0.070 | 0.103 | 0.059 | 0.714 | −0.021 | 0.103 | 0.058 | 0.665 |

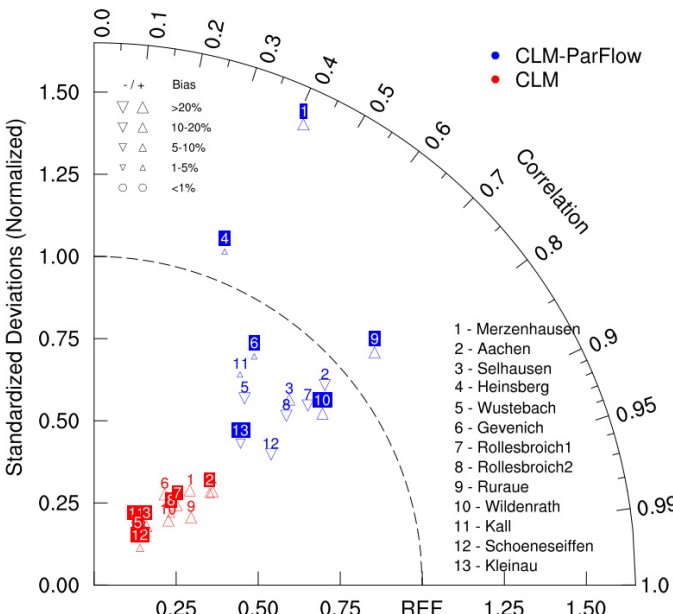

**Figure 3.** Taylor diagram of the soil moisture of CLM and CLM-ParFlow model runs as compared to 13 CRNS observations.

After overlooking systematic bias, ubRMSD for the CLM stand-alone model and the coupled model are not significantly different and both below 0.06 cm$^3$ cm$^{-3}$, which indicates that both simulations could produce reasonable results. The correlation between CRNS observations and land surface simulations ranges from 0.576 to 0.821 for CLM stand-alone and 0.365 to 0.805 for CLM-ParFlow. Both CLM and CLM-ParFlow simulations show a strong correlation with in situ data, suggesting that the soil moisture dynamics at 500 m scale can be relatively well captured by both models.

### 3.3. Temporal and Spatial Correlation between Model Simulations and the SMAP L3_SM_P_E Product

Figure 4 shows daily mean soil moisture content in the upper soil layer (5 cm) over the research area for CLM and CLM-ParFlow simulations compared with the SMAP L3_SM_P_E product, together with daily precipitation time series. The SMAP product and model simulations have different soil wetting and drying dynamics after rainfall. The near-surface soil is sensitive to precipitation because of intensive positive vertical water gradients, but tends to dry quickly after water infiltration related to evaporation [86]. Although the representation of L-band sensing depth at 5 cm has been using for remote sensing validations [26,87,88]; however, the L-band sensing depths are affected by soil moisture, and the penetration depth can be shallower when soil moisture is high [89]. The penetration depths and vertical soil moisture gradients lead to different drying behavior. The simulations match soil moisture from SMAP well during most of the year. Generally, CLM overestimates the soil water content during summer. Compared to SMAP, CLM-ParFlow simulations show a very small wet bias of only 0.004 cm$^3$/cm$^3$ while CLM has a larger wet bias of 0.065 cm$^3$ cm$^{-3}$. The RMSD for CLM and CLM-ParFlow simulations are 0.085 cm$^3$ cm$^{-3}$ and 0.045 cm$^3$ cm$^{-3}$ respectively.

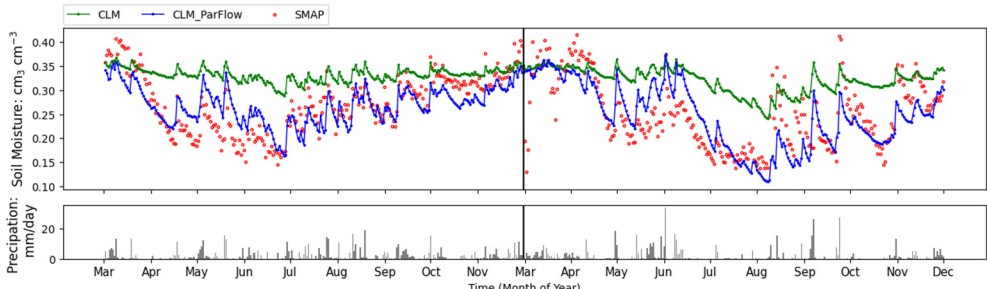

**Figure 4.** Soil moisture time series (0–5 cm) averaged over the simulation domain for 2017 and 2018 from CLM and CLM-ParFlow simulations compared with SMAP product.

Spatial maps of performance indices are given in Figure 5 and illustrate the differences between model predictions and observations. For the maps at 500 m resolution, as the resolution of the simulations is finer than the satellite measurement, large differences between simulation and measurements occur in valleys and rivers regions because these areas are not well covered by the coarser resolution of the SMAP satellite.

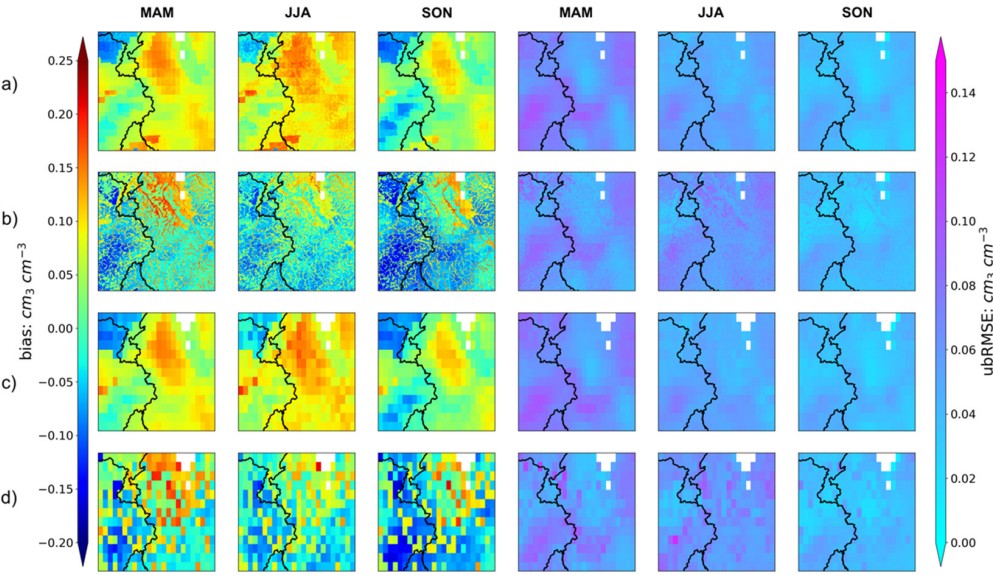

**Figure 5.** Spatial maps of bias (left three columns) and ubRMSD (right three columns) of soil moisture between SMAP and CLM model (**a,c**), SMAP and CLM-ParFlow model (**b,d**) for MAM, JJA and SON over the investigation area posted at (upper) the resolution of the model grid (250 m) and (lower) at the resolution of SMAP (9 km).

In most of the area, CLM has a higher soil water content than SMAP. The CLM-ParFlow model captures the spatial variability of soil water content and shows the influence of the river network. The soil moisture is close to saturation in the river valley. Meanwhile, the soil is drier in hilly areas. This is related to the difference in subsurface process representation between CLM and CLM-ParFlow. In Parflow, the Richards' equation is used to calculate 3D subsurface water flow, including both vertical and lateral water movement which includes the lateral groundwater flow which moves water from the hilly areas towards the river valleys. In addition, lateral flow by streams and rivers is also modeled. The water flow convergence process (i.e., the lateral redistribution of water via streams and aquifers from hills to the river valleys and lowlands) can be better captured by CLM-ParFlow than by CLM, which just considers vertical water flow. In the northern flat and valley areas, the differences of soil moisture between CLM and CLM-ParFlow is smaller, where precipitation and infiltration excess is low, thus the later water flow redistribution processes have smaller impact.

Further minor discrepancies between CLM and CLM-ParFlow simulations are related to the different estimation of soil hydraulic parameters (different pedo-transfer functions). Several areas where CLM shows larger deviations with respect to SMAP soil moisture coincide with loam regions of the soil texture map, which indicates that for loam texture CLM is probably too wet, which could be related to the default pedo-transfer function in CLM, which might underestimate hydraulic conductivity for loamy soil types. The spatial distribution of both bias and ubRMSD also show clear differences between the northern flat area (with larger bias) and the southern hilly area (smaller bias). This is probably related to the fact that precipitation is high in the hills and both CLM and SMAP have soil moisture values close to field capacity. Concerning CLM-ParFlow, the differences with SMAP are larger in the Rhine valley, ParFlow overestimates the influence of the river on soil moisture values in the areas next to the stream. Notice that the CLM-ParFlow model is not calibrated, and that river-groundwater interaction could be closer to real conditions by adjusting for example riverbed hydraulic conductivities. Both bias and ubRMSD values show that the discrepancies between model simulations and SMAP data are smallest in autumn for CLM simulations and CLM-ParFlow.

### 3.4. Triple Collocation

In general, the statistics of a time series triple is inherently unique, so that a comparison of different triples in a collocation analysis is not directly possible. However, the same SMAP product and CRNS data are used in this study to be compared to two different simulation results. That provides a large ability for a direct comparison. Table 7 shows the TC results for model simulations (as $Y$) and SMAP L3_SM_E_P product (as $Z$) compared to reference CRNS datasets (as $X$). Per definition, the scaling factor $\beta_X$ of the reference dataset is 1, while the SMAP product and model simulations are scaled to this reference product. $\beta_y$ and $\beta_z$ values larger than 1 indicate that the dynamic range of the datasets from the SMAP product and the model simulations is lower than that of CNRS soil moisture time series, and vice versa. For CLM stand-alone simulation, the average $\beta_y$ is 4.538 and $\beta_z$ is 1.370. For CLM-ParFlow, $\beta_y$ and $\beta_z$ are quite close (1.453 and 1.242 separately). The scaling factors larger than 1 indicate that both SMAP L3_SM_P_E and the land surface model simulations have underestimated the soil moisture dynamics at the CRNS-sites, and probably in the complete research area. Compared to CLM stand-alone simulations, the $\beta_y$ of the coupled model is closer to 1, indicating less need to scale. In terms of $\beta_z$, it should be noted that the scaling factors for Wüstebach and Wildenrath are lower than for other stations, indicating that the dynamics range of retrieved soil moisture needs to be reduced to match the scale of CRNS observations and model simulation time series. For these two stations, some additional problems (see Section 3.1) seem to influence the soil moisture retrieval results. Although the soil moisture datasets used in the TC correspond to different depths (SMAP for the upper 5 cm and the other two datasets for the upper 20 cm), no obvious relationship between the penetration depth differences and soil moisture dynamics is detected. Table 7 also provides the absolute TC error standard deviation. The ranges for $\sigma_{\epsilon Y}$ are 0.010 to 0.026 (CLM) and 0.028 to 0.080 (CLM-ParFlow) respectively. In general, the SMAP soil moisture product provides similar error standard deviations with $\sigma_{\epsilon Z} \leq 0.060$ in both triples.

**Table 7.** Triple collocation comparison results for model simulations (as *Y*) and SMAP L3_SM_E_P product (as *Z*) compared to reference CRNS datasets (as *X*).

| Name | Z: CLM | | | | | | Z: CLM-ParFlow | | | | | |
|---|---|---|---|---|---|---|---|---|---|---|---|---|
| | $\sigma_{\epsilon X}$ | $\sigma_{\epsilon Y}$ | $\sigma_{\epsilon Z}$ | $\beta_Y$ | $\beta_Z$ | $SNR_Y$ | $\sigma_{\epsilon X}$ | $\sigma_{\epsilon Y}$ | $\sigma_{\epsilon Z}$ | $\beta_Y$ | $\beta_Z$ | $SNR_Y$ |
| Merzenhausen | 0.016 | 0.024 | 0.061 | 3.015 | 1.285 | −1.397 | 0.021 | 0.080 | 0.056 | 1.134 | 1.088 | −7.456 |
| Aachen | 0.020 | 0.025 | 0.057 | 2.610 | 1.270 | 1.485 | 0.023 | 0.037 | 0.050 | 1.199 | 1.025 | 6.576 |
| Selhausen | 0.017 | 0.015 | 0.061 | 5.669 | 1.556 | −2.082 | 0.011 | 0.045 | 0.059 | 1.415 | 1.441 | 2.622 |
| Heinsberg | 0.013 | 0.026 | 0.056 | 2.346 | 1.391 | 3.351 | 0.015 | 0.060 | 0.054 | 1.980 | 1.283 | −11.020 |
| Wüstebach | 0.028 | 0.008 | 0.055 | 5.941 | 0.783 | 1.330 | 0.028 | 0.036 | 0.055 | 1.526 | 0.780 | −1.414 |
| Gevenich | 0.020 | 0.024 | 0.060 | 3.520 | 1.376 | −3.166 | 0.040 | 0.041 | 0.052 | 1.257 | 1.052 | 4.035 |
| Rollesbroich1 | 0.023 | 0.024 | 0.058 | 3.698 | 1.455 | −1.250 | 0.030 | 0.038 | 0.050 | 1.267 | 1.144 | 8.143 |
| Rollesbroich2 | 0.019 | 0.023 | 0.059 | 3.858 | 1.579 | −0.489 | 0.039 | 0.038 | 0.050 | 1.322 | 1.192 | 8.099 |
| Ruraue | 0.008 | 0.019 | 0.065 | 2.806 | 1.691 | 6.315 | 0.038 | 0.058 | 0.068 | 1.383 | 2.164 | 0.826 |
| Wildenrath | 0.018 | 0.010 | 0.057 | 3.689 | 0.889 | 4.258 | 0.021 | 0.028 | 0.056 | 1.072 | 0.831 | 7.878 |
| Kall | 0.006 | 0.017 | 0.049 | 7.141 | 1.443 | −6.570 | 0.023 | 0.058 | 0.046 | 1.825 | 1.337 | −4.374 |
| Schoeneseiffen | 0.011 | 0.010 | 0.061 | 7.090 | 1.429 | 2.879 | 0.010 | 0.036 | 0.060 | 1.774 | 1.383 | 4.907 |
| Kleinhau | 0.026 | 0.014 | 0.061 | 7.607 | 1.664 | −2.779 | 0.023 | 0.040 | 0.057 | 1.730 | 1.424 | 3.800 |
| Average | 0.017 | 0.018 | 0.058 | 4.538 | 1.370 | 0.145 | 0.025 | 0.046 | 0.055 | 1.453 | 1.242 | 1.740 |

The linearized *SNR* value gives the ratio relationship between soil moisture and uncertainty magnitude. On average, the $SNR_Y$ of CLM and CLM-ParFlow are 0.145 and 1.740 dB respectively. The negative $SNR_Y$ values demonstrate that the random noise is larger than the soil moisture signal. In the CLM-ParFlow simulation, most sites show a better performance than the CLM model. As described before, Ruraue is along the river, where the soil moisture is sensitive to river parameterization in ParFlow. Heinsberg, Kall, and Kleinhau have large negative values, indicating that both models have large absolute errors. Overall, the CLM-ParFlow could provide more valuable results than the stand-alone model.

*3.5. Effect of Lateral Water Flow on Soil Moisture*

Previous research indicated that the lateral flow is important in land surface modeling, especially when the resolution is fine [90]. The spatial patterns of modeled soil moisture show that the CLM-ParFlow has wet grid cells at foothills and valleys. The soil moisture gradient is larger in the wet grid cells and surrounding drier grid cells. By taking account of lateral flow, soil moisture decreases in these wet areas due to lateral diffusion. Also, the lateral drainage driven by topographic gradient results in soil moisture redistribution. In view of previous studies [91], lateral flow is expected for steep hillocks, even if slight difference in soil texture between adjoining grid cells. Also, the accumulated runoff in ParFlow, generated by infiltration excess or saturation excess, can route or reinfiltrate, while some other traditional LSMs can remove excess water from modeled water cycle [92], CLM-ParFlow maintains high soil moisture in convergence areas. This confirms the importance of lateral subsurface flow on the hydrological cycle, especially in mountainous areas.

**4. Conclusions**

This study compared soil moisture data from cosmic ray neutron sensors (CRNS), passive microwave remote sensing (SMAP L3_SM_E_P product) and land surface model simulations by the Community Land Model (CLM, version 3.5) and the coupled land surface-subsurface model CLM-ParFlow over a 150 × 150 km region in western Germany. CLM-ParFlow can better capture the impact of groundwater on soil moisture than CLM as it has a more advanced subsurface physical process scheme. With this approach an analysis of the impact of the representation of subsurface processes in hydrological simulations of soil moisture was performed. The evaluation results can be summarized as follows:

Over 13 CRNS sites, the SMAP L3_SM_E_P product shows a small bias of 0.005 cm$^3$ cm−$^3$ only compared to the CRNS observations. Nevertheless, local differences can be large (up to −0.120 cm$^3$ cm−$^3$ for the densely forested Wüstebach site) due to differing

spatial resolution of the soil moisture products and errors in soil texture and land cover used to derive soil moisture from brightness temperature in the SMAP L3_SM_E_P product. Besides, the disturbing role of dense vegetation and complex topographic features influence the accuracy of the SMAP product. Overall, the unbiased root mean square error (ubRMSE) is around 0.056 cm$^3$ cm$-^3$, indicating that SMAP L3_SM_E_P product could barely meet its mission requirement for this very heterogeneous and hilly region.

The comparison between CRNS and land surface simulations show that CLM has a wet bias (0.070 cm$^3$ cm$^{-3}$) and CLM-ParFlow has a dry bias ($-0.021$ cm$^3$ cm$^{-3}$). Local biases can be large, which might be related to the uncertainty in soil texture and hydraulic conductivity, inadequate pedotransfer functions and lack of consideration for soil bulk density in CLM model. In terms of ubRMSE, both CLM and CLM-ParFlow are below 0.06 cm$^3$/cm$^3$ and compare well to CRNS observation dynamics. The SMAP product and CLM-ParFlow do not show a systematic difference in soil moisture, which is in contrast to most land surface models which are wetter than SMAP.

The triple collocation (TC) comparison implies that both CLM and CLM-ParFlow show similar noise levels with $\sigma_{\epsilon Z}$ below 0.058. The scaling factor of CLM-ParFlow is less than a third of CLM stand-alone, indicating that the coupled model could perform better with respect to CRNS measurements. This is an important aspect for future data assimilation studies, as the typical adaptation of the soil moisture climatology of model and observation becomes less mandatory. The higher SNR (signal-to-noise ratio) value for the coupled model CLM-ParFlow also indicates it can provide more valuable results than the CLM stand-alone model.

It should be noted that the direct metrics (e.g., RMSE and *r*) do not show a clear better performance of the CLM-ParFlow model compared to the CLM stand-alone model. The TC method shows that the simulation has been improved when lateral subsurface dynamics is involved. Unlike typical performance metrics, where the assumption is that the reference data set is free of (random) errors, TC methods account for sensor and representativeness errors and can be considered more robust than conventional metrics and close to reality [45]. With conventional evaluation metrics, we focus on the dynamics of the different time series instead of the absolute soil moisture values, because there can be a systematic bias between CRNS and SMAP measurements, as well as model simulations, which is related to different underlying assumptions for the different measurement and simulation methods. This method is also used because it is standard in the land surface modeling literature and allows therefore an easier comparison with other papers [36,40,93,94].

In summary, the model structure is important for soil moisture modeling. Compared to CLM-ParFlow model, the CLM model has a simplified representation of describing the soil moisture variability while neglecting lateral water flow. The CLM model tends to overestimate the soil moisture and provided similar soil moisture estimation in grid cells that have the same soil type and plant functional type. The lateral subsurface process in CLM-ParFlow lead to soil water redistribution and improvements in prediction. The coupled model can describe the spatial variability of soil moisture. It is worth to consider lateral subsurface flow in LSMs to have more accurate soil moisture simulation.

However, some limitations should be noted. First, lateral subsurface flow takes mainly place in the saturated subsurface. However, there is also evidence for lateral flow in the top layer of the unsaturated zone for sloping soil due to rainfall dynamics. Nevertheless, the CRNS measurements provide in the first place only information on soil moisture, and are less suited to evaluate how well the influence of groundwater is represented by models. Second, the CRNS measurements might be slightly biased, caused by the limited number of observation sites, scale mismatches, and imperfect calibration. The bias is simply set-aside when using the same statistical evaluation methods in order to compare these results with other remote sensing and land surface modeling studies. Finally, this study only covers three seasons for the years of 2017 (quite average conditions) and 2018 (very dry). A longer time may be desirable to better evaluate the relative performance of the model, including

different weather conditions. Also, a finer soil map resolution and larger study domain would be desirable in future studies.

**Author Contributions:** Conceptualization, H.Z., C.M., R.B, H.V., and H.-J.H.F.; Methodology, C.M. and H.-J.H.F.; Formal analysis, H.Z.; Investigation, H.Z.; Writing—original draft preparation, H.Z.; Writing—review and editing, H.Z., C.M., R.B., H.V., and H.-J.H.F.; Supervision, C.M. and H.-J.H.F. All authors have read and agreed to the published version of the manuscript.

**Funding:** This research was funded by China Scholarship Council (CSC), grant number. 201806010358.

**Acknowledgments:** We thank the supercomputing center of Forschungszentrum Jülich for their computational support and our access to the JURECA and JUWELS supercomputer; we thank the Terrestrial Environmental Observatories (TERENO) community for providing observation data for free; we are gratefully to the three anonymous reviewers and editor for helping us improve the quality of this paper. H.Z. acknowledges the financial support from the China Scholarship Council (CSC) and support from Dragon 5 Cooperation between ESA and the Ministry of Science and Technology(MOST) of the P.R.China under the grant ID.59316; C.M. acknowledges support from the German Federal Ministry of Economics and Technology for the AssimEO project under the grant 50EE1914A. R.B. acknowledges funding from the European Union's Horizon 2020 research and innovation program under grant agreement No 871128 (eLTER PLUS).

**Conflicts of Interest:** The authors declare no conflict of interest.

## Appendix A

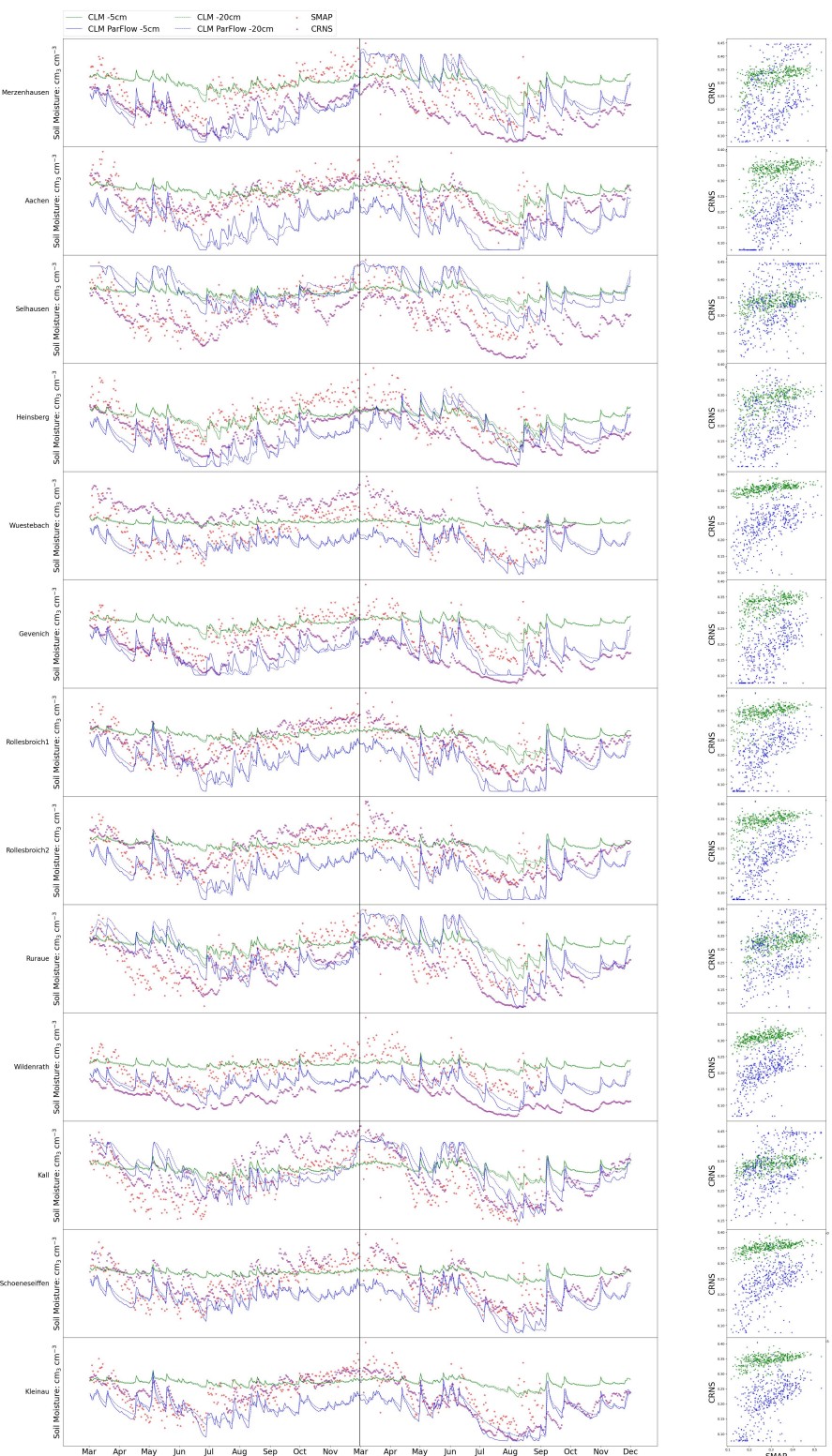

**Figure A1.** Time series and scatterplots of soil moisture from SMAP (red), CRNS (purple) and LSMs CLM at 5 cm (green solid) and at 20 cm (green dashed); CLM-ParFlow at 5 cm (blue solid) and at 20 cm (blue dashed) at 13 sites.

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
