# Peer review of "The Importance of Subsurface Processes in Land Surface Modeling over a Temperate Region: An Analysis with SMAP, Cosmic Ray Neutron Sensing and Triple Collocation Analysis"

_remotesensing, doi:10.3390/rs13163068_

Round 1

Reviewer 1 Report

The paper by Zhao et al. evaluates two versions of a land surface model, one with a traditional 1-D soil hydrology scheme (CLM) and the other explicitly considering the 3-D water flow in the subsurface (CLM-ParFlow), in terms of their performance in simulating soil moisture dynamics in a temperate climate region near western Germany. The model results are compared with the site-level CRNS soil moisture observations and the satellite-derived SMAP soil moisture product, using both conventional performance metrics and the Triple Collocation (TC) method. Results of the TC analysis show a better consistency of CLM-ParFlow than that of CLM compared with the CRNS observations (lower scaling factors and higher SNR values), thus highlighting the importance of the representation of subsurface hydrological processes in land surface models, especially at high resolutions. Overall, the paper is well written and logically organized. But I have some comments that need to be addressed before publication.

First, the evaluation result using conventional metrics like RMSD and Pearson’s r does not show a better performance of CLM-ParFlow (and even slightly lower r for CLM-ParFlow than for CLM). A bit more discussion is needed regarding the different results between conventional metrics and the TC method, and some perspectives/insights about why the TC method may be more appropriate would add more value to this paper.

Second, about the model configuration, is a two-year spinup for CLM (Line 217) long enough for soil hydrology? Why doesn’t the CLM-ParFlow simulation do a spinup like CLM, but instead start from an initial condition from a previous study? Could this different treatment lead to different soil moisture results by the two models already? Besides, it would be helpful to specify in the Methods the additional inputs (if any) required by CLM-ParFlow than CLM.

Finally, I suggest adding a Taylor diagram which displays the evaluation metrics for SMAP, CLM, and CLM-ParFlow. This could provide a more concise visual summary than the current two Tables 4 and 6. Also, time-series plots of the sites (in an Appendix perhaps), similar to the nice Figure 3, would be very helpful to provide readers a more intuitive sense of the model performances.

Specific comments:

Abstract: the main result is not made clear. Currently it only writes in the last sentence “Coupled and uncoupled model performed differently at individual evaluation CRNS and catchment-wide”, which is not a good summary of the results.

Figure 2: Spell out NRW in the figure caption. In (b), is it correct that there are quite some broadleaf evergreen tropical trees (PFT2)? Besides, there are many shrubs (PFT5) in the figure, which seems inconsistent with the description at Line 199-200.

Line 191: ? is porosity? Please check.

Line 222: equation 4, the weights of each soil layer do not sum up to one.

Figure 3: please enlarge the texts.

Figure 4: in the caption: “left/right two columns” should be “three columns”; “upper/lower” should be specified as “a,b” and “c,d”. Besides, the rainbow color scheme is perceptually misleading, please consider changing the color scheme.

References: Some refs are repeated in the numbering, for example 51 and 53, 45 and 68, 23 and 37. In addition, some refs are not relevant to the sentence where they are cited, for example the refs cited at Line 162. Please check throughout the text.

Author Response

Dear Reviewer,

thank you for your detailed summary and suggestions.

We gladly incorporated these suggestions in the revised manuscript and corrected all minor inconsistencies.

Please find our responses in the attachment.

Reviewer 2 Report

The manuscript is well structured and nicely presented on an important topic - the importance of representing the subsurface physics in LSM. The discussion is extensive and informative, and generally answers almost all questions that the reader wants to ask. I am happy with the manuscript in its present form.

Author Response

Dear Reviewer,

thank you very much!

Reviewer 3 Report

see attached file

Author Response

(The authors gave the same response as above.)
